# An Evaluation of Psychological Well-Being among Physicians and Nurses in Makkah's Major Hospitals

**Huda Alghamdi** [1,*] , **Abdullah Almalki** [2] **and Maha Alshaikh** [1]

1    Department of Mental Health, King Abdulaziz Hospital, Makkah 24221, Saudi Arabia
2    Department of Quality Management, King Abdulaziz Hospital, Makkah 24221, Saudi Arabia
\*    Correspondence: hohalghamdi@gmail.com

**Abstract:** Background: Physicians and nurses experience poor psychological well-being relative to other employees in healthcare fields. This study aimed to evaluate the psychological well-being among physicians and nurses in Makkah's major hospitals. Methods: In this cross-sectional study, 460 physicians and nurses from seven major hospitals in Makkah were recruited to investigate their psychological well-being using the General Health Questionnaire-12 (GHQ-12) based on social dysfunction, anxiety, and confidence loss. Results: Over half (64.3%) of the physicians and nurses in this study scored at or above the GHQ-12 cut-off point (12), which is a positive result for poor psychological well-being. There were significant differences in the psychological well-being mean between Saudis and non-Saudis ($t = 2.203$, $p = 0.028$), years of work experience ($t = 3.349$, $p = 0.001$), hospitals (F = 2.848, $p = 0.010$), attending psychological support sessions ($t = 2.082$, $p = 0.038$), and history of visiting psychological clinics ($t = -4.949$, $p < 0.001$). There was also a significant association between the three GHQ-12 factors and the participants' socio-demographic characteristics. Conclusion: The psychological well-being of physicians and nurses is low. The alarming number of physicians and nurses suffering from social dysfunction, anxiety, and loss of confidence should be addressed in Makkah's major hospitals. The employee assistance program (EAP) could be highly valuable and effective for addressing the well-being of employees and their personal problems that may impact their work performance, conduct, health, and overall well-being at the Ministry of Health.

**Keywords:** psychological health; Saudi Arabia; general health; GHQ-12; employee assistance program (EAP)

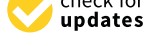



## 1. Introduction

Psychological health is an important factor that contributes to people's overall well-being. According to the World Health Organization (WHO), psychological health is a state of well-being in which each individual knows their own potential, copes with life's everyday stresses, works productively, and contributes to their community [1]. Psychological health is a balanced state of physical, mental, and social well-being, not just the absence of negative issues [2]. The General Health Questionnaire-12 (GHQ-12) is unique and one of the most extensively used self-report instruments for evaluating psychological stress and disorders. Measuring social dysfunction, anxiety, and confidence loss can successfully gauge an individual's level of well-being [3]. Various studies have been conducted using the GHQ, including population-based studies and employee health assessment surveys [4]. In recent years, low levels of psychological well-being have received more attention. The best method for treating people with personal issues, unwanted behavior, and addiction at work is to make an employee assistance program (EAP) available [5]. According to Nunes et al. [6], the main goals of these programs are to address existing problems and promote healthy living among employees. Unwell physicians and nurses could compromise healthcare quality and safety. To achieve the quadruple objective in healthcare, it is essential for healthcare systems to invest in infrastructure that delivers evidence-based

treatments that foster a culture that supports clinician health and well-being. The National Academy of Medicine (NAM) developed a model that pinpoints elements that influence clinician well-being and resiliency. External and individual components were discovered in this model. The regulatory, business, payer, and learning and practice environments, as well as organizational factors, are examples of external influences [7]. The healthcare role, personal factors, talents, and capacities are examples of individual factors. A wide range of interventions, especially within the individual components mentioned in the model, have been developed to increase physicians' well-being. The NAM has produced a website containing an information hub to promote medical professionals' well-being and resourcefulness, drawing on numerous resources [7]. In Saudi Arabia, two empirical studies of physicians and nurses conducted by Alosaimi et al. [8,9] revealed that an average of 22.5% of 935 participants experienced stress. The findings also revealed that perceived stress correlated with many demographic variables, such as whether the physician or nurse is a Saudi national, female or male, the magnitude of their work, and the occurrence of fatigue and sleep deprivation. The surveyed consultants (85% of the target sample) reported experiencing work-related problems, such as work, academic, and life stress. In addition, over 50% of participants contemplated changing their work and working abroad to relieve the considerable stress [8,9]. Job demands-resources (JD-R) is the most prevalent occupational health psychology model for examining the relationship between job characteristics and employee well-being. The JD-R model implies two causal processes: health impairment and motivation. Burnout and negative outcomes, such as health complaints and turnover intentions, are increased when job demands are high. In contrast, job resources serve as motivation, stimulate work engagement, and lead to positive outcomes within an organization, including performance and commitment. Moreover, job resources can buffer the impact of burnout due to job demands. However, a lack of job resources contributes to burnout [10]. Health staff face substantial work pressure, and they need ample counselling to ensure that their work remains unaffected by work-related stress. This study illustrated the level of psychological well-being of physicians and nurses, and it highly recommended establishing a specialized EAP that concentrates mainly on health-sector employees in Makkah's major hospitals.

## 2. Materials and Methods

This cross-sectional study was conducted among 460 physicians and nurses working in seven major hospitals in Makkah City, Saudi Arabia. The hospitals included Alnoor Hospital, Maternity Hospital, King Faisal Hospital, Hera Hospital, Ajyad Hospital, King Abdul-Aziz Hospital, and King Abdullah Medical City; together, these hospitals will be hereafter referred to as the Makkah Health Cluster. All the hospitals are public sector hospitals, managed by the Ministry of Health, Saudi Arabia, and located in Makkah, with a total bed capacity 2694 beds (between 300 to 500 beds in each hospital). It serves around 1,700,000 citizens, residents, and pilgrims of Makkah city and the surrounding area. Makkah is the third-most populated city in Saudi Arabia after Riyadh and Jeddah. It is situated 70 km from Jeddah on the Red Sea. According to the Health Ministry's 2018 statistical yearbook, the Makkah Health Cluster includes 3796 physicians and 6910 nurses [11]. A cluster random sampling technique was used to select the participants from each hospital. Participants were limited to physicians and nurses who worked in the Makkah region hospitals, regardless of their nationalities, genders, ages, positions, educational levels, and cultural backgrounds. Other professionals, such as administrative staff and retired physicians, were excluded.

The sample size was calculated using the Raosoft sample size calculator (http://www.raosoft.com/samplesize.html (accessed on 16 May 2021)), with a 5% margin of error and a 95% confidence interval. Based on the total population, a minimum sample size of 460 participants was required for this study.

Data were collected from June to November 2021, with an overall sample size of 460 employees (physicians and nurses) from the Makkah Health Cluster. The participants'

psychological well-being was measured using the 12-item version of the General Health Questionnaire (GHQ-12). This is a self-administered screening tool designed for diagnosing mental disturbances and disorders [12]. It is also capable of detecting healthcare employees who are likely to have (or are at high risk of developing) psychiatric disorders. It consists of three factors: social dysfunction, anxiety, and loss of confidence. The GHQ has been translated into 38 different languages, including Arabic, due to its reliability and validity [13,14]. The Arabic version of the GHQ-12 proved to be reliable, as indicated by the Cronbach alpha of 0.86. The best balance between sensitivity and specificity was found at the General Health Questionnaire cut-off point of 15/16: at this threshold, sensitivity was 0.88, and was paired with a specificity of 0.84 [14]. The questionnaire was sent via online channels, such as email and other social communication channels.

The GHQ-12 contains an equal number of positive and negative items, with each item scored using a Likert scale ranging from 0 to 3. For positive items, the response options consist of 'better than usual' (0), 'same as usual' (1), 'worse than usual' (2), and 'much worse than usual' (3). For negative items, the response options consist of 'not at all' (0), 'less than usual' (1), 'same as usual' (2), and 'more than usual' (3). The total score ranges from 0 to 36, with scores of 12 and above indicating poor psychological well-being [14]. The survey also includes questions about demographic information, such as gender, experience level, nationality, profession, hospital name, department, and whether the respondent has attended any psychological support sessions or visited any psychological clinics.

The data were analyzed using the Statistical Package for the Social Sciences (SPSS), version 25 (Chicago, IL, USA). Descriptive statistics were used to analyze and summarize the data. Means $\pm$ standard deviations (SDs) and ranges were used to describe continuous data, whereas frequencies and percentages were used to describe categorical data. All data were normally distributed prior to any statistical tests. Independent *t*-tests or one-way ANOVAs, along with the Tukey–Kramer method, were used to determine the mean GHQ-12 scores, the significant differences between groups, and the associations between socio-demographic characteristics and GHQ-12 factors. The significance level was set at $p = 0.05$.

The study was granted ethical approval by the local institutional review board, the General Directorate of Research, Makkah Health Cluster, Saudi Arabia. Research participants voluntarily participated in this study, and each participant provided written informed consent. The data collected from the participants were kept confidential.

## 3. Results

In the present study, 457 participants out of 460 were included for data analysis purposes (a response rate of 99.3%). Table 1 summarizes the respondents' socio-demographic characteristics. Briefly, male participants represented 49.2% of the study population, and female participants represented 50.8%. Saudis constituted 63.7% of the sample, whereas non-Saudis made up 36.3%. The majority of the participants (*n* = 270, 59.1%) were nurses. Most (74.8%) of the respondents had ≤15 years of experience. A total of seven major hospitals participated in this study. The highest percentage (28.7%) of responses came from Alnoor Hospital, followed by King Abdul-Aziz Hospital (20.4%). The participants worked in different departments, including medical wards (21.4%), allied health departments (9.5%), and outpatient departments (8.5%). Most participants had not received psychological support (92.8%) or visited any psychological clinics (87.7%).

**Table 1.** Distribution of Socio-demographic Characteristics of the Participants (N = 457).

| Variables | Frequency | Percentage (%) |
|---|---|---|
| **Gender** | | |
| Male | 225 | 49.2 |
| Female | 232 | 50.8 |
| **Nationality** | | |
| Saudi | 291 | 63.7 |
| Non-Saudi | 166 | 36.3 |
| **Profession** | | |
| Physician | 187 | 40.9 |
| Nurse | 270 | 59.1 |
| **Experience, Mean (SD)** | 12.04 (7.73) | |
| ≤15 years | 342 | 74.8 |
| >15 Years | 115 | 25.2 |
| **Department** | | |
| Surgical wards | 73 | 16.0 |
| Medical wards | 98 | 21.4 |
| Emergency department | 85 | 18.6 |
| Intensive care unit (ICU) | 73 | 16.0 |
| Outpatient department | 39 | 8.5 |
| Allied health services department | 89 | 19.5 |
| **Hospital name** | | |
| Alnoor | 131 | 28.7 |
| Maternity | 54 | 11.8 |
| King Faisal | 50 | 10.9 |
| Hera Hospital | 61 | 13.3 |
| Ajyad | 17 | 3.7 |
| King Abdul-Aziz | 93 | 20.4 |
| King Abdullah Medical City | 51 | 11.2 |
| **Have you attended any psychological support sessions?** | | |
| Yes | 33 | 7.2 |
| No | 424 | 92.8 |
| **Have you visited any psychological clinics?** | | |
| Yes | 56 | 12.3 |
| No | 401 | 87.7 |

SD = Standard Deviation.

In this study, the total GHQ-12 scores ranged from 2 to 34, with a mean (SD) score of 14.95 (6.40)—far higher than the cut-off point of 12. Among the respondents, 64.3% had scores of ≥12, indicating poor mental health and well-being (Figure 1). The levels of psychological well-being of physicians and nurses across various groups are shown in Table 2. The GHQ-12 comprises six positive and six negative items to assess positive and negative mental health; in particular, the highest average scores were for items 5 (under stress), 7 (enjoy the day-to-day activities), and 8 (face up to problems). The mean (SD) score for item 5 was 1.79 (0.98). The majority of respondents (61.3%) scored 2 or 3 points for this item; only 10.9% of respondents scored 0, indicating that the majority of respondents constantly felt under stress. The mean (SD) score for item 7 was 1.62 (0.88). About half (50.3%) of the respondents scored 2 or 3 points for this item; and only 7.7% of respondents scored 0, showing that, in general, the respondents did not enjoy their day-to-day activities. The mean (SD) score for item 8 was 1.60 (0.77). Most (57.6%) respondents scored 2 or 3 points for this item; only 7.2% of respondents scored 0, showing that most of the respondents face up to problems.

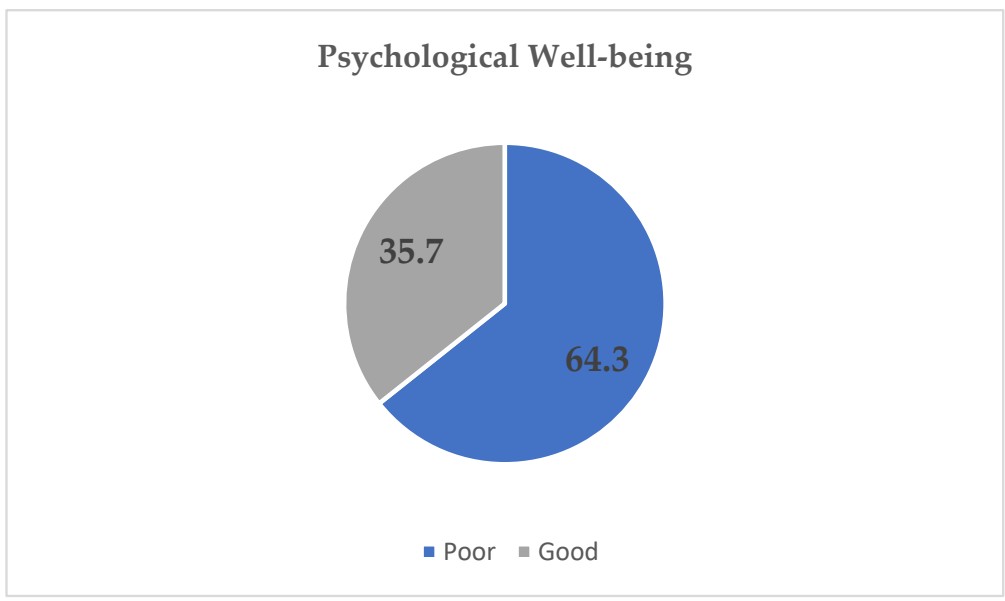

**Figure 1.** Proportion of psychological well-being among the participants.

**Table 2.** Distribution of GHQ-12 items and summary scores (N = 457).

| | Questionnaire Item | Mean (SD [a]) | Type of Item | * Response Frequencies (%) | | | |
|---|---|---|---|---|---|---|---|
| | | | | 0 | 1 | 2 | 3 |
| GHQ-1 | Able to concentrate | 1.14 (0.77) | Positive | 86 (18.8) | 243 (53.2) | 107 (23.4) | 21 (4.6) |
| GHQ-2 | Lost much sleep | 1.49 (1.02) | Negative | 90 (19.7) | 146 (31.9) | 129 (28.2) | 92 (20.1) |
| GHQ-3 | Playing a useful part | 1.11 (0.80) | Positive | 91 (19.9) | 259 (56.7) | 73 (16.0) | 34 (7.4) |
| GHQ-4 | Capable of making decisions | 1.02 (0.73) | Positive | 100 (21.9) | 269 (58.9) | 69 (15.1) | 19 (4.2) |
| GHQ-5 | Under stress | 1.79 (0.98) | Negative | 50 (10.9) | 127 (27.8) | 149 (32.6) | 131 (28.7) |
| GHQ-6 | Could not overcome difficulties | 1.12 (0.80) | Negative | 98 (21.4) | 233 (51.0) | 101 (22.1) | 25 (5.5) |
| GHQ-7 | Enjoy your day-to-day activities | 1.62 (0.88) | Positive | 35 (7.7) | 192 (42.0) | 143 (31.3) | 87 (19.0) |
| GHQ-8 | Face up to problems | 1.60 (0.77) | Positive | 33 (7.2) | 161 (35.2) | 217 (47.5) | 46 (10.1) |
| GHQ-9 | Feeling unhappy and depressed | 1.34 (1.057) | Negative | 121 (26.5) | 140 (30.6) | 114 (24.9) | 82 (17.9) |
| GHQ-10 | Losing confidence | 0.75 (0.91) | Negative | 236 (51.6) | 119 (26.0) | 80 (17.5) | 22 (4.8) |
| GHQ-11 | Thinking of self as worthless | 0.59 (0.88) | Negative | 284 (62.1) | 101 (22.1) | 48 (10.5) | 24 (5.3) |
| GHQ-12 | Feeling reasonably happy | 1.39 (0.84) | Positive | 52 (11.4) | 228 (49.9) | 124 (27.1) | 53 (11.6) |
| | **Mean GHQ-12 score [b]** | **14.95 (6.40)** | **Range (2–34)** | | | | |

* Note: Likert method coding as zero representing most healthy, and 3 representing poor healthy; [a] SD, standard deviation; [b] a higher score indicates a worse situation.

Independent *t*-tests and one-way ANOVA tests were conducted to determine the relationships between the participants' psychological well-being scores and their socio-demographic characteristics (Table 3). There were significant differences in the mean psychological well-being scores between Saudis and non-Saudis ($t$ = 2.203, $p$ = 0.028), those with ≤15 years and >15 years of work experience ($t$ = 3.349, $p$ = 0.001), hospitals (F = 2.848, $p$ = 0.010), those who had and had not attended psychological support sessions (t = 2.082, $p$ = 0.038), and those who had and had not visited psychological clinics ($t$ = −4.949, $p$ < 0.001). However, there were no significant differences in the mean psychological well-being scores based on gender, profession, or department.

**Table 3.** Distribution of psychological well-being among participants across socio-demographic variables.

| Variables | Mean | (SD) | Test Statistic | |
| --- | --- | --- | --- | --- |
| | | | *t*/F | *p*-Value |
| **Gender** | | | | |
| Male | 15.10 | 6.43 | 0.443 [a] | 0.658 |
| Female | 14.82 | 6.39 | | |
| **Nationality** | | | | |
| Saudi | 15.45 | 6.48 | 2.203 [a] | 0.028 * |
| Non-Saudi | 14.10 | 6.21 | | |
| **Profession** | | | | |
| Physician | 15.50 | 6.27 | 1.467 [a] | 0.143 |
| Nurse | 14.59 | 6.49 | | |
| **Experience** | | | | |
| ≤15 years | 15.54 | 6.45 | 3.349 [a] | 0.001 * |
| >15 Years | 13.24 | 5.99 | | |
| **Department** | | | | |
| Surgical wards | 14.47 | 6.39 | 0.949 [b] | 0.449 |
| Medical wards | 15.74 | 6.41 | | |
| Emergency department | 17.74 | 7.06 | | |
| Intensive care unit (ICU) | 13.95 | 4.83 | | |
| Outpatient department | 12.82 | 5.96 | | |
| Allied health services department | 14.25 | 6.95 | | |
| **Hospital name** | 15.88 | 5.81 | | |
| Alnoor | | | 2.848 [b] | 0.010 * |
| Maternity | 14.97 | 5.64 | | |
| King Faisal | 15.97 | 6.56 | | |
| Hera Hospital | 15.27 | 7.29 | | |
| Ajyad | 14.67 | 6.67 | | |
| King Abdul-Aziz | 14.05 | 5.50 | | |
| King Abdullah Medical City | 14.18 | 6.05 | | |
| **Have you attended any psychological support sessions?** | | | | |
| Yes | 12.82 | 5.74 | 2.082 [a] | 0.038 * |
| No | 15.13 | 6.43 | | |
| **Have you visited any psychological clinics?** | | | | |
| Yes | 18.82 | 6.28 | −4.949 [a] | <0.001 * |
| No | 14.42 | 6.24 | | |

SD = Standard deviation; [a] independent T test; [b] one-way ANOVA; * significant at *p* < 0.05

Table 4 provides a comparative analysis of the mean differences in the psychological well-being levels between the hospitals, which shows the following statistical data: King Faisal vs. Alnoor—the poor mental health levels in the King Faisal participants are higher than the Alnoor participants, with a significant mean difference of 3.29; King Faisal vs. Hera Hospital—the poor mental health levels in the King Faisal participants are higher than the Hera Hospital participants, with a significant mean difference of 3.79; and King Faisal vs. King Abdul-Aziz—the poor mental health levels in the King Faisal participants are higher than the King Abdul-Aziz participants, with a significant mean difference of 3.49.

Table 5 provides a summary of the association between all three factors in the GHQ—social dysfunction, anxiety/depression, and loss of confidence—as well as the socio-demographic variables. There were significant associations between social dysfunction factors and gender ($t = 2.284$, $p = 0.023$), nationality ($t = 3.165$, $p = 0.002$), years of experience ($t = 2.427$, $p = 0.016$), attendance at psychological support sessions (F = 2.960, $p = 0.003$), and history of visiting psychological clinics (F = −3.156, $p = 0.002$). In addition, there were significant associations between anxiety/depression factors and years of experience ($t = 3.265$, $p = 0.001$), department (F = 2.667, $p = 0.022$), hospital name (F = 2.662, $p = 0.015$), and history of visiting psychological clinics ($t = −4.808$, $p < 0.001$). Years of experience

($t = 2.985$, $p = 0.003$), hospital (F = 3.133, $p = 0.005$), and history of visiting psychological clinics ($t = -5.106$, $p < 0.001$) were significantly associated with loss-of-confidence factors.

**Table 4.** Multiple comparisons of the psychological well-being levels between hospitals with Tukey HSD post-hoc test.

| (I) Hospital Name | (J) Hospital Name | Mean Difference (I-J) | 95% Confidence Interval | | *p*-Value |
|---|---|---|---|---|---|
| | | | Lower Bound | Upper Bound | |
| Alnoor | Maternity | −1.29036 | −4.3199 | 1.7392 | 0.869 |
| | King Faisal | −3.28962 * | −6.4038 | −0.1754 | 0.031 |
| | Hera hospital | 0.49956 | −2.4043 | 3.4034 | 0.999 |
| | Ajyad | 1.62685 | −3.2026 | 6.4563 | 0.954 |
| | King Abdul-Aziz | 0.20307 | −2.3372 | 2.7433 | 1.000 |
| | King Abdullah Medical City | −1.43197 | −4.5240 | 1.6601 | 0.817 |
| Maternity | Alnoor | 1.29036 | −1.7392 | 4.3199 | 0.869 |
| | King Faisal | −1.99926 | −5.6760 | 1.6775 | 0.676 |
| | Hera hospital | 1.78992 | −1.7105 | 5.2903 | 0.736 |
| | Ajyad | 2.91721 | −2.2928 | 8.1272 | 0.644 |
| | King Abdul-Aziz | 1.49343 | −1.7117 | 4.6986 | 0.812 |
| | King Abdullah Medical City | −0.14161 | −3.7996 | 3.5164 | 1.000 |
| King Faisal | Alnoor | 3.28962 * | 0.1754 | 6.4038 | 0.031 |
| | Maternity | 1.99926 | −1.6775 | 5.6760 | 0.676 |
| | Hera hospital | 3.78918 * | 0.2153 | 7.3631 | 0.030 |
| | Ajyad | 4.91647 | −0.3432 | 10.1761 | 0.084 |
| | King Abdul-Aziz | 3.49269 * | 0.2074 | 6.7780 | 0.029 |
| | King Abdullah Medical City | 1.85765 | −1.8707 | 5.5860 | 0.759 |
| Hera hospital | Alnoor | −0.49956 | −3.4034 | 2.4043 | 0.999 |
| | Maternity | −1.78992 | −5.2903 | 1.7105 | 0.736 |
| | King Faisal | −3.78918 * | −7.3631 | −0.2153 | 0.030 |
| | Ajyad | 1.12729 | −4.0106 | 6.2652 | 0.995 |
| | King Abdul-Aziz | −0.29649 | −3.3831 | 2.7901 | 1.000 |
| | King Abdullah Medical City | −1.93153 | −5.4861 | 1.6230 | 0.676 |
| Ajyad | Alnoor | −1.62685 | −6.4563 | 3.2026 | 0.954 |
| | Maternity | −2.91721 | −8.1272 | 2.2928 | 0.644 |
| | King Faisal | −4.91647 | −10.1761 | 0.3432 | 0.084 |
| | Hera hospital | −1.12729 | −6.2652 | 4.0106 | 0.995 |
| | King Abdul-Aziz | −1.42378 | −6.3653 | 3.5177 | 0.979 |
| | King Abdullah Medical City | −3.05882 | −8.3054 | 2.1877 | 0.598 |
| King Abdul-Aziz | Alnoor | −0.20307 | −2.7433 | 2.3372 | 1.000 |
| | Maternity | −1.49343 | −4.6986 | 1.7117 | 0.812 |
| | King Faisal | −3.49269 * | −6.7780 | −0.2074 | 0.029 |
| | Hera hospital | 0.29649 | −2.7901 | 3.3831 | 1.000 |
| | Ajyad | 1.42378 | −3.5177 | 6.3653 | 0.979 |
| | King Abdullah Medical City | −1.63504 | −4.8993 | 1.6292 | 0.755 |
| King Abdullah Medical City | Alnoor | 1.43197 | −1.6601 | 4.5240 | 0.817 |
| | Maternity | 0.14161 | −3.5164 | 3.7996 | 1.000 |
| | King Faisal | −1.85765 | −5.5860 | 1.8707 | 0.759 |
| | Hera hospital | 1.93153 | −1.6230 | 5.4861 | 0.676 |
| | Ajyad | 3.05882 | −2.1877 | 8.3054 | 0.598 |
| | King Abdul-Aziz | 1.63504 | −1.6292 | 4.8993 | 0.755 |

* The mean difference is significant at the 0.05 level.

**Table 5.** The association between socio-demographic characteristics with GHQ-12 factors (social dysfunction factor, anxiety/depression factor, and loss of confidence factor).

| Variables | Social Dysfunction Factor | | | Anxiety/Depression Factor | | | Loss of Confidence Factor | | |
|---|---|---|---|---|---|---|---|---|---|
| | *t*/F | DF | *p* Value | *t*/F | DF | *p* Value | *t*/F | DF | *p* Value |
| Gender | 2.284 | 455 | 0.023 * | −0.800 | 455 | 0.424 | −0.524 | 455 | 0.600 |
| Profession | 1.900 | 455 | 0.058 | 1.829 | 455 | 0.068 | −0.760 | 455 | 0.448 |
| Nationality | 3.165 | 455 | 0.002 * | 1.266 | 455 | 0.206 | 1.058 | 455 | 0.291 |
| Experience | 2.427 | 455 | 0.016 * | 3.265 | 455 | 0.001 * | 2.985 | 455 | 0.003 * |
| Department | 1.990 | 5 | 0.079 | 2.667 | 5 | 0.022 * | 1.114 | 5 | 0.352 |
| Hospital | 2.037 | 6 | 0.059 | 2.662 | 6 | 0.015 * | 3.133 | 6 | 0.005 * |
| Have you attended any psychological support sessions? | 2.960 | 455 | 0.003 * | 1.875 | 455 | 0.061 | −0.181 | 455 | 0.856 |
| Have you visited any psychological clinics? | −3.156 | 455 | 0.002 * | −4.808 | 455 | <0.001 * | −5.106 | 455 | <0.001 * |
| Variables | 2.284 | 455 | 0.023 * | −0.800 | 455 | 0.424 | −0.524 | 455 | 0.600 |

\* Significant at *p* < 0.05.

## 4. Discussion

The present study evaluated the levels of psychological well-being among 457 physicians and nurses at major hospitals in Makkah, Saudi Arabia, to determine whether an assistance program (such as EAP) would be necessary. In this study, 64.3% of the participants scored at or above the GHQ-12 cut-off point of 12, indicating that physicians and nurses in Saudi Arabia experience high levels of psychological distress. The direct comparison of the levels of psychological well-being in our study with previous measurements among physicians and nurses in Saudi Arabia is difficult due to a lack of published studies examining psychological well-being using the GHQ-12. However, our findings were similar to those from previous studies among physician and nurse populations [15–18].

In our study, the GHQ-12 included three factors based on a previous literature review; the first was labelled as social dysfunction (items 1, 3, 4, 7, 8, and 12), the second was labelled as anxiety (items 2, 5, 6, and 9), and the third was labelled as loss of confidence (items 10 and 11). As shown in Table 2, the social dysfunction and anxiety factors loadings were high, whereas the loss-of-confidence factor was low. These results indicate that most participants suffered from social dysfunction and anxiety. This study's factor structure of the GHQ-12 was similar to those of several other international versions [14,19–22].

Some previous studies have investigated psychological distress among physicians and nurses in Saudi Arabia; however, we believe that our study may be the first to assess levels of psychological well-being using the GHQ-12 instrument. Comparing our results with those from studies among physicians and nurses in Saudi Arabia that used different instruments revealed that the psychological distress levels in our study were higher than those in some studies [22,23], whereas they were similar to those in other studies [24,25]. Although physicians and male respondents reported higher scores for psychological distress than nurses and female respondents, these differences were not significant. This result is similar to that from a study by Rutledge et al. [26], in which the differences in psychological well-being scores between professions and genders were not significant. This may be due to the fact that both physicians and nurses, regardless of gender, provide similar medical services.

Non-Saudi participants reported higher psychological well-being scores than Saudis in the present study; this difference was significant (*p* = 0.028). In contrast, previous studies did not find significant differences in the mean psychological well-being scores between participants of different nationalities or races [16,24]. In the present study, a significant difference in psychological well-being was also identified based on the participants' experience levels; those with ≤15 years of experience scored higher for psychological distress than those with >15 years of experience. Similarly, a previous study reported significantly different mean psychological well-being scores based on the years of experience of physi-

cians and nurses [25]. However, other studies reported no significant differences [18]. More experience may play a significant role in developing the capacities necessary to deal with work-related stress, and, thus, increases psychological well-being.

The present study reported no significant differences in the mean psychological well-being scores across different departments of the hospitals. This contrasts with the results from a study by Seaman et al. [11], in which physicians and nurses in intensive care units showed the highest levels of psychological distress. These higher levels may be associated with the stressful responsibilities of dealing with patients who suffer from acute symptoms, and need high levels of care. These problems adversely affect individual clinicians' well-being.

The results indicated that there was a statistically significant difference between the mean psychological well-being scores of participants with a history of visiting psychological clinics and the scores of those who did not attend any psychological support sessions. The participants with a history of visiting psychological clinics and those who did not attend any psychological support sessions reported higher scores of psychological distress. This result was supported by Melnyk et al.'s study [27], which focused on randomized controlled trials that tested psychological support sessions to improve the mental health, well-being, physical health, and lifestyle behaviors of physicians and nurses. Their results indicated that mindfulness and cognitive-behavioral-therapy-based interventions effectively reduced stress, anxiety, and depression. Brief interventions that incorporate deep breathing and gratitude may be beneficial. Visual triggers, pedometers, and health coaching with texting increase physical activity [27]. Participants in King Faisal Hospital in Makkah were found to be at high risk of psychological distress. They had a higher mean score (17.74) for psychological distress than those in other hospitals, and these differences were significant ($p$ = 0.010). Many factors, including work stress because of the large number of patients or the pressure of management, can explain this difference between the hospitals in this study.

This study found that there were significant associations between social dysfunction factors and gender, nationality, experience, attendance at psychological support sessions, and a history of visiting psychological clinics. Furthermore, there were significant associations between anxiety/depression factors and experience, department, hospital name, and a history of visiting psychological clinics. There were also significant associations between loss-of-confidence factors and experience level, hospital name, and a history of visiting psychological clinics. These findings correspond with those from previous studies [17,23,25,28].

In terms of limitations, as we conducted our study in Makkah and not in other regions of Saudi Arabia, generalizing our results to the whole country may be of slight difficulty. However, the characteristics of the population of Makkah are similar to those of the entire country, which is a mix of urban and rural residents. The number of physician participants in this study was lower than that of nurse participants; however, this is unsurprising because the number of nurses is higher in hospitals. Additionally, the cross-sectional design only indicated associations between risk factors, not causation, limiting the generalization of our findings. Finally, collecting data using self-reported questionnaires is susceptible to response bias.

## 5. Conclusions

In conclusion, this study found low levels of psychological well-being among the participants. Most participants felt constantly under stress, did not enjoy their day-to-day activities, and faced up to problems. There were significant differences in the mean psychological well-being between Saudis and non-Saudis, years of work experience, hospitals, attending psychological support sessions, and a history of psychological clinic visits. Therefore, this study recommends the establishment of well-being clinics in all Saudi cities to treat psychological disorders, considering the systematic procedures and policies of the Ministry of Health. It also recommends using EAP as an online tool to assess and evaluate the psychological well-being of Ministry of Health employees confidentially and

flexibly. EAP will help health staff understand or overcome their difficulties, regardless of the source is work, or otherwise, through specialized counselling services and awareness programs under the management of mental health specialists and doctors to maintain their productivity and safeguard their psychological status, so that they can deal with their psychological distress such as fatigue, work stress, and others. and others. In addition, a future study on the causes and treatment of psychological distress among physicians and nurses at hospitals in Saudi Arabia is recommended.

**Author Contributions:** Conceptualization, H.A. and M.A.; data analysis, H.A. and A.A.; data curation, A.A.; writing—original draft preparation, H.A.; writing—review and editing, H.A., M.A. and A.A.; project administration, H.A. All authors have read and agreed to the published version of the manuscript.

**Funding:** This research received no external funding.

**Institutional Review Board Statement:** This study was conducted in accordance with the Declaration of Helsinki, and approved by the local institutional review board, the General Directorate of Research, Makkah Health Cluster, Saudi Arabia (H-02-K-076-0621-523).

**Informed Consent Statement:** Informed consent was obtained from all subjects involved in the study.

**Data Availability Statement:** The datasets analyzed/generated during the current study are available from the corresponding author on reasonable request.

**Acknowledgments:** We wish to thank all the physicians and nurses who participated in this study. The authors appreciate the directors of the hospitals for their consent and assistance in completing the data collection for this study.

**Conflicts of Interest:** The authors declare no conflict of interest.

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
