# Peer review of "An Evaluation of Psychological Well-Being among Physicians and Nurses in Makkah’s Major Hospitals"

_2571-8800, doi:10.3390/j5030025_

Round 1
Reviewer 1 Report
This study assessed psychological well-being using the GHQ-12 among doctors and nurses working in major hospitals in Makkah. Psychological well-being was shown to be low overall and to vary according to background factors. From the findings, it was concluded that the well-being of doctors and nurses needs to be improved, suggesting the importance of future programme to improve the well-being of employees and providing important issues to consider when considering mental health.
However, the following points should be considered in the publication.
(1) Although significant differences in psychological well-being were found in hospitals, Table 2 shows that significant differences were found in departments and not in hospitals. Based on the results in Table 2, we believe that the results and discussion section should be re-examined.
(2) The seven main hospitals in Makkah are covered, but it would be easier to understand if the number of beds and other information were included as an indication of the size of each hospital.
3) In the conclusion section, it would be desirable to have specific suggestions on how to utilise the Employee Assistance Programme.
Author Response
Response to Reviewer 1 Comments
Point 1: Although significant differences in psychological well-being were found in hospitals, Table 2 shows that significant differences were found in departments and not in hospitals. Based on the results in Table 2, we believe that the results and discussion section should be re-examined.
Response 1: The results and discussion section were re-examined. The significant differences in psychological well-being were found in hospitals and table 2 was corrected accordingly.
Point 2: The seven main hospitals in Makkah are covered, but it would be easier to understand if the number of beds and other information were included as an indication of the size of each hospital.
Response 2: Summary of the hospitals and the study location were included. Details of each hospital are not available.
Point 2: In the conclusion section, it would be desirable to have specific suggestions on how to utilise the Employee Assistance Programme.
Response 3: We included how to utilise the Employee Assistance Programme in the conclusion section

Reviewer 2 Report
It is about a topic of interest in that the impact of the pandemic on frontline staff is being investigated,
I have some recommendations for revising the manuscript, detailed below:
(1) It is desirable that the introduction describes the theoretical framework in which the research design, hypotheses, etc. will fit. I suggest you consult the job-resources-demands model from occupational psychology
(2) The psychometric qualities of the Arabic version of the questionnaire used in the research should be mentioned
(3) In the ANOVA analysis, you must also add post hoc analysis
(4) It would be interesting to separately analyze the correlation between stress, respectively anxiety and the socio-demographic variables in addition to what you have already calculated.
(5) It is only a suggestion, considering that the triad of depression, stress and anxiety is emblematic of mental health. You will be able to report what you have obtained with previous studies on this topic - depression, stress, anxiety in a pandemic context.
Good luck to the authors in revising the manuscript
Author Response
Response to Reviewer 2 Comments
Point 1: It is desirable that the introduction describes the theoretical framework in which the research design, hypotheses, etc. will fit. I suggest you consult the job-resources-demands model from occupational psychology
Response 1: We include information regarding the job-resources-demands model from occupational psychology in the introduction.
Point 2: The psychometric qualities of the Arabic version of the questionnaire used in the research should be mentioned.
Response 2: The psychometric qualities of the Arabic version of the questionnaire used in the research were included.
Point 3: In the ANOVA analysis, you must also add post hoc analysis
Response 3: We added post hoc analysis in Table 4.
Point 4: It would be interesting to separately analyze the correlation between stress, respectively anxiety and the socio-demographic variables in addition to what you have already calculated.
Response 4:: Yes, I agree with you, but this paper aims to evaluate the psychological state of the physicians and nurses in order to determine if the CAP is necessary. In addition, if we include all of these Tables will exceed the limited tables counts.
Point 5: It is only a suggestion, considering that the triad of depression, stress and anxiety is emblematic of mental health. You will be able to report what you have obtained with previous studies on this topic - depression, stress, anxiety in a pandemic context.
Response 5: Yes, I agree with you, but this paper aims to evaluate the psychological state of the physicians and nurses in order to determine if the CAP is necessary. We are not focusing on depression, stress and anxiety specifically.
